# 3 versus 7 Tesla magnetic resonance imaging for parcellations of subcortical brain structures in clinical settings

**Bethany R. Isaacs**[1,2], **Martijn J. Mulder**[1,3], **Josephine M. Groot**[1], **Nikita van Berendonk**[1], **Nicky Lute**[1,4], **Pierre-Louis Bazin**[1,5], **Birte U. Forstmann**[1☯], **Anneke Alkemade**[1☯] *

1 University of Amsterdam, Integrative Model-Based Cognitive Neuroscience Research Unit, Amsterdam, The Netherlands, 2 Department of Experimental Neurosurgery, Maastricht University Medical Centre, Maastricht, The Netherlands, 3 Psychology and Social Sciences, University of Utrecht, Utrecht, The Netherlands, 4 Clinical Neuropsychology, Vrije University, Amsterdam, The Netherlands, 5 Max Planck Institute for Human, Cognitive and Brain Sciences, Leipzig, Germany

☯ These authors contributed equally to this work.
* jmalkemade@gmail.com

**Data Availability Statement:** All anonymized data and analysis scripts are available from https://osf. io/4nrku/, under the terms of the Creative Commons Attribution License and complies with

## Abstract

7 Tesla (7T) magnetic resonance imaging holds great promise for improved visualization of the human brain for clinical purposes. To assess whether 7T is superior regarding localization procedures of small brain structures, we compared manual parcellations of the red nucleus, subthalamic nucleus, substantia nigra, globus pallidus interna and externa. These parcellations were created on a commonly used clinical anisotropic clinical 3T with an optimized isotropic (o)3T and standard 7T scan. The clinical 3T MRI scans did not allow delineation of an anatomically plausible structure due to its limited spatial resolution. o3T and 7T parcellations were directly compared. We found that 7T outperformed the o3T MRI as reflected by higher Dice scores, which were used as a measurement of interrater agreement for manual parcellations on quantitative susceptibility maps. This increase in agreement was associated with higher contrast to noise ratios for smaller structures, but not for the larger globus pallidus segments. Additionally, control-analyses were performed to account for potential biases in manual parcellations by assessing semi-automatic parcellations. These results showed a higher consistency for structure volumes for 7T compared to optimized 3T which illustrates the importance of the use of isotropic voxels for 3D visualization of the surgical target area. Together these results indicate that 7T outperforms c3T as well as o3T given the constraints of a clinical setting.

## Introduction

The availability of 7 Tesla (T) Magnetic Resonance Imaging (MRI) scanners has rapidly increased in recent years [1–3]. The theoretical benefits of anatomical 7T MRI over lower field strengths can be attributed to the increased spatial resolution, contrast- and signal-to-noise ratios (CNR and SNR, respectively), which collectively result in higher quality imaging within

the rules of the General Data Protection Regulation (EU) 2016/679.

**Funding:** This research was supported by a Vidi and Vici grant from the Dutch Organization for Scientific Research (BUF), and an NWO-STW grant from the Dutch Organization for Scientific Research (BUF, MJM, AA).

**Competing interests:** The authors have declared that no competing interests exist.

feasible time frames [4, 5]. Improved visibility of pathological alterations on 7T has been reported in the literature for brain tumors [6], epilepsy [7], multiple sclerosis [8], stroke [9], and neurodegenerative diseases [10]. However, to what extent increased visibility afforded by 7T has the potential to improve clinical outcomes regarding invasive neuro interventions remains unknown.

A promising clinical application of 7T MRI is the target visualization of structures for deep brain stimulation (DBS) surgery [1, 11]. DBS procedures target structures within the subcortex, which is comprised of a large number of small, iron and calcium-rich nuclei that are located in close proximity to one another [2]. The main DBS targets for PD and dystonic disorders are the globus pallidus interna (GPi) and subthalamic nucleus (STN) [12–15]. Identification of the STN benefits from visualization of the border of the SN, which has also been targeted for epilepsy [16]. Also the parcellation of the GPi benefits from visualizing the boundary with the external segment of the GP (GPe), the stimulation of which has been shown to modulate functional connectivity in Huntington's disease patients [17]. Additionally, the red nucleus (RN) is often used as a landmark for identification and orientation of the surrounding nuclei [18].

Alterations in biometals such as iron in human tissue are commonly observed in pathological processes, for instance, the occurrence of dopaminergic neurodegeneration of the substantia nigra (SN) in Parkinson's disease (PD). Such changes in the chemical composition can cause disease specific structural alterations in shape, volume and location [19–21]. Moreover, the neurophysical properties of both physiological and aberrant accumulation of biometals can be exploited to increase the visibility of structural boundaries with both ultra-high field (UHF) MRI and tailored post-processing techniques, such as quantitative susceptibility mapping (QSM) [22–25].

Conventional MRI can fail to capture the detailed local neuroanatomy due a weaker field strength, resulting in reduced spatial resolution, signal and contrast. These limitations can be directly translated into a clinical setting with regards to the accuracy of MRI based targeting protocols for DBS implantations. DBS of the STN has been related to a number of psychiatric, cognitive, and emotional disturbances [26]. Moreover, a fraction of patients will fail to respond to stimulation and or maintain their parkinsonian symptoms, and may require the removal or reimplantation of their DBS leads [26, 27]. These failures to appropriately respond to neurointervention can partially be attributed to suboptimal placement of the DBS lead as a consequence of both inaccurate visualization of the target and reliance on landmark identification [28]. Additionally, DBS surgeries commonly incorporate intra-operative micro electrode recordings and behavioral testing in awake patients to confirm optimal lead placement [29–31]. This is a time-consuming procedure and distressing for the patient. The higher spatial accuracy that 7T MRI offers could contribute to more accurate surgical targeting and clinical efficacy. Additionally, it can reduce the length of the surgery and the requirement for reimplantation, while ultimately contributing to the abolishment of the need for awake testing during surgery and dramatically improving patient comfort [1, 32].

Clinical MRI often includes parallel imaging (PI) techniques to reduce acquisition time which is associated with an SNR penalty. This is warranted for both practical reasons, to improve image contrast, as well as clinical reasons, as patients with movement disorders cannot be scanned for extended periods of time. PI reconstructions result in spatially varying noise amplification, which is reflected in the g-factor. However, PI can result in both g-factor penalties and longitudinal magnetization saturation, which can produce anatomically inaccurate and distorted images [2, 33]. In clinical practice, anisotropic voxel sizes are commonly employed in order to maintain a higher SNR in-plane.

In our current studies, we investigate the potential of 7T for improved targeting with a quantitative comparison of 3T with 7T MRI scans. We acquired two sets of 3T data; one

representative of the resolution of clinical 3T (c3T) MRI typically used for DBS targeting, as well as an optimized set of 3T (o3T). Additionally, we obtained a set of 7T data from the same participants. We would like to clarify that we could not run the same optimized protocol at 3T and 7T. Running the 7T protocol at 3T would result in an unacceptably increased scan time at 3T which would preclude clinical implementation. Furthermore, the increase in specific-absorption-rates (SAR) escalating magnetization would result in local tissue heating, thereby posing a severe health risk to those scanned. Direct quantitative comparisons were drawn from both manual and semi-automated parcellations the GPe, GPi, RN, SN, and STN. Given the iron rich nature of these deep brain structures, and our previous studies indicating that for such structures QSM outperforms $T2^*$-weighted images we used QSM contrasts for parcellations [34]. Additionally, a semi-automated parcellation approach was employed to parcellate the GP, RN, SN and STN, in order to identify potential biases occurring with manual parcellations and whether accuracy increases with field strength.

## Methods

### Participants

10 healthy participants (male = 2, female = 8, mean age = 25.9 y, S.D age = 5.8 y), healthy as assessed by self-reports, were scanned at the Spinoza Centre for Neuroimaging in Amsterdam, The Netherlands, on a Philips 7T and 3T Achieva MRI system, with a 32-channel head array coil. The research was approved in writing by the LAB Ethics Review Board of the Faculty of Social and Behavioral Sciences, the local Ethical Committee of the Department of Psychology at the University of Amsterdam (ERB number 2016-DP-6897). All participants provided written informed consent prior to the scanning, and structural 7T MRI data was included in the Amsterdam ultra-high field adult lifespan database (AHEAD) [35].

### Data acquisition

**c3 Tesla.** Whole-brain T1-weighted images obtained with a 3D Turbo/Fast Field Echo (TFE) sequence with 1mm isotropic voxel sizes, field of view (FOV) = 240 x 188, 220 slices, echo time (TE) = 3.7 ms, repetition time (TR) = 8.2 ms, TFE factor = 142, TFE shots = 118, $SENSE_{PA}$ = 2.5, acquisition time (TA) = 04:42 min, obtained in the transverse plane. Whole-brain T2-weighted images obtained with a Turbo/Fast Spin Echo sequence (TSE) with 0.45 x 0.45 x 2mm voxel sizes, FOV = 230 x 182, 48 slices, TE = 80 ms. TR = 3000 ms, TSE factor = 15, TSE shots = 150, TA = 06:12 min, obtained in the transverse plane. Total acquisition time was 10:54 min.

**o3 Tesla.** Whole-brain T1-weighted images were obtained with a 3D Fast Field Echo (FFE) sequence with 1mm isotropic voxel sizes, FOV = 240 x 188, 220 slices, TE = 3.7 ms, TR = 8.2 ms, TFE factor = 142, TFE shots = 293, TA = 11:38 min in the transverse plane (no acceleration factor). Whole brain T2-weighted images were acquired with 3D Fast Field Echo (FFE) sequence with voxel sizes 1mm isotropic, TE1, 2, 3, 4, 5 = [4.1, 9.8, 13.85, 19.55, 23.60 ms], TR = 46 ms, echo spacing (ES) = 9.75 ms, FA = 20, FOV = 240 x 188, 140 slices, $SENSE_{PA}$ = 2, TA = 10:08 min. The main difference between the clinical and optimized 3T scans is the voxel size. Two separate scans were collected with o3T, with a total acquisition time of 21:46 min. We would like to note that we were unable to match the o3T spatial resolution with that of the 7T due to specific absorption rate (SAR) limitations.

**7 Tesla.** For 7T, one scan incorporating both T1 and $T2^*$ contrasts was obtained using a MP2RAGEME (magnetization-prepared rapid gradient echo multi-echo) sequence [36]. The MP2RAGEME is an extension of the MP2RAGE sequence by [37] and consists of two rapid gradient echo $(GRE_{1,2})$ images that are acquired in sagittal plane after a 180° degrees inversion

pulse and excitation pulses with inversion times $TI_{1,2}$ = [670 ms, 3675.4 ms]. A multi-echo readout was added to the second inversion at four echo times ($TE_1$ = 3ms, $TE_{2,1-4}$ = 3, 11.5, 19, 28.5 ms). Other scan parameters include flip angles $FA_{1,2}$ = [4˚, 4˚]; $TR_{GRE1,2}$ = [6.2 ms, 31 ms]; bandwidth = 404.9 MHz; $TR_{MP2RAGE}$ = 6778ms; acceleration factor $SENSE_{PA}$ = 2; FOV = 205 x 205 x 164 mm; acquired voxel size = 0.70 x 0.7 x 0.7 mm; acquisition matrix was 292 x 290; reconstructed voxel size = 0.64 x 0.64 x 0.7 mm; Turbo/Fast factor (TFE) = 150 resulting in 176 shots; Total acquisition time = 19:53 min.

## Image calculations

$T2^*$ maps for o3T and 7T MRI scans were created by least-squares fitting of the exponential signal decay over the multi-echo images of the second inversion. 7T T1-weighted images were generated by complex ratio of the product of first and second inversion over the sum of their square [37]. A quantitative T1 map was also reconstructed from this T1-weighted image via a look-up table procedure [37]. For QSM, the 3T data underwent more extensive clipping at the frontal and sinus regions as compared to the 7T MRI data. This was required since the algorithm is sensitive to non-local artefacts, which are more prominent in these regions on o3T MRI scans. For QSM, phase maps were pre-processed using iHARPERELLA (integrated phase unwrapping and background phase removal using the Laplacian) of which the QSM images were computed using LSQR [38, 39]. Scans were reoriented and skull information was removed as described previously [40]. The c3T MRI sequence did not allow the calculation of quantitative $T2^*$ maps or QSM images.

## Parcellation methods

**Manual parcellation.** Inspection of the c3T scans revealed that despite the high in-plane resolution, which allowed the identification of the structures of interest in the axial plane, we were unable to create a biologically plausible 3-dimensional reconstruction of the structures of interest due to the anisotropic nature of the voxel sizes. We therefore decided not to pursue further analyses of the c3T MRI scans. Multi-echo data was not acquired, and therefore it was not possible to reconstruct QSM images for parcellations.

For o3T and 7T images, manual parcellations were performed in individual space using the QSM images for the GPe/i, RN, SN, and STN by two independent trained researchers. Given the level of familiarity of these raters with MRI data, we concluded that blinding for the scan sequence was impossible. T1-maps and/or T1-weighted images were used for additional anatomical orientation and identification of landmarks such as the ventricles, pons and corpus callosum. $T2^*$-maps were also used where appropriate. See S1 File for the approach used for manual parcellations. Raters were blind to each other's parcellations, and inter-rater agreement was determined by the Dice correlation coefficient (see statistical methods).

*Semi-automated parcellation*: *Multimodal Image Segmentation Tool (MIST)*. Semi-automated parcellation was performed for the combined GPe/i, RN, SN and STN with FSL's Multimodal Image Segmentation Tool (MIST) [41, 42]. QSM-maps and T1-weighted images were used as input for MIST. MIST output parcellations were compared across field strength (o3T vs 7T), as well as across parcellation method (manual vs. semi-automated) in order to assess for potential biases in manual parcellations such as order or practice effects.

The o3T brain extracted T1-weighted and QSM maps were co-registered via a multi-step process, where first whole brain $T2^*$-maps were registered to the corresponding T1-weighted images using FLIRT (as implemented in FSL version 6.0.1) with 6 degrees of freedom, nearest neighbor interpolation and mutual information cost function. This transformation was then applied to the QSM-maps, extrapolated form the fifth echo of the $T2^*$ sequence, also with 6

degrees of freedom, mutual information cost function and instead a sinc interpolation. The same transforms were applied to the manual parcellations to allow for direct comparisons with MIST outputs. All registrations were visually inspected for misalignments by comparing the following landmarks: ventricles, pons, and corpus callosum.

The 7T MP2RAGEME sequence allowed the calculation of all contrasts from a single sequence and thus in the same space, not requiring any registration steps. The MP2RAGE was used as the whole-brain anatomical reference image and the fourth echo of the second inversion was used for the $T2^*$ image. Resampling was achieved with Nibabel (version 2.3), with second order spline interpolation, and constant mode parameter. Where appropriate, the header information was copied from the fourth echo of the second inversion to the MP2RAGE. Images were resampled as MIST only handles (near) isotropic voxel sizes. MIST was unable to perform parcellations in 0.7 mm isotropic voxels, which we attributed to the limited information provided by the prior derived from MNI-space for these small voxels. Images were therefore resampled to 0.8 mm which resolved the problem.

## Dice coefficients

Dice coefficients were assessed to determine interrater reliability [43]. Dice scores were compared between $o$3T and 7T images to test the directed hypothesis that 7T images result in higher interrater agreement as compared to $o$3T images. The Dice coefficient was calculated as follows:

$$Dice = \frac{2 \times \lor m1 \cap m2 \lor}{m1 \lor + m2 \lor}$$

Where $|m_i|$ is the size of mask $i$ and $|m_1 \cap m_2|$ is the size of the conjunct mask of mask 1 and 2. A conjunct mask of a set of masks M only includes voxels included by both raters [43].

## Volume calculations

For manual parcellations, all volume calculations were performed using the conjunct volume of the individual raters, as described previously. Calculations for manual Dice factors were calculated in the space in which the parcellations were performed [34, 44]. Masks from the MIST output were compared with manually parcellated conjunction masks resampled to 0.8mm for the 7T data, and the masks that were registered from $T2^*$ to T1 for the $o$3T MRI data.

## Anatomical distance

The anatomical distance between the centers of mass of the individual structures in the left and right hemisphere was assessed, providing a measure for changes in the perceived location of the individual structures across field strengths. We expected that altered visibility of specific anatomical borders would be reflected in a bilateral shift in the center of mass and as a result an altered anatomical distance.

Distances were calculated as follows:

$$Distance(l, r) = \sum_{i=1}^{n}$$

$l$ and $r$ correspond to the left and right hemisphere. The square root of the sum is obtained by adding the power of the left $x$, $y$ and $z$ coordinates of the center of mass of the individual structures $l$ and $r$ [45].

## Contrast to noise ratios

Contrast-to-noise ratios (CNRs) of the QSM images were calculated to assess differences in visibility of the anatomical structure under investigation. Intensities of non-zero voxels were extracted using the *segmentation_statistics* implemented in Nighres [46]. The CNR was calculated as follows:

$$CNR = \frac{S_{I-S_O}}{\sigma_O}$$

$S_I$ is the signal inside the mask, represented by the mean value of all the voxels in the conjunct mask. $S_O$ is the signal outside the mask, calculated as the mean value of all voxels that directly border the outside of the disjunct mask (all voxels scored inside the mask of a single rater). $\sigma_0$ is the standard deviation of the set of QSM intensities in these voxels. This approach was adopted to ensure that voxels outside the mask were not part of the separate individual masks.

## Statistical methods

All statistical analyses were conducted within a Bayesian framework (Table 1) using the Bayes-Factor toolbox [47] in R [48], interpreted in light of the assumptions proposed by [49] and adapted by [50]. Bayesian approaches draw their distribution from the observed data, rather than requiring a Gaussian distribution, which can be difficult to meet in both small and or clinical datasets when using a frequentist perspective. Bayesian statistics sample from the data itself, the priors for each group are calculated independently, in which smaller groups will receive greater uncertainty compared to a larger group for comparison, with no trade off with statistical power [51]. Therefore, a limited number of subjects should not negatively impact our results [52, 53]. Each test is performed independently of the others so we assume multiple comparisons are not a confounder in the present study [54–56]. We incorporated a within subjects' approach, and for all analyses data was collapsed across hemisphere.

**Manual o3T v 7T.** For manual parcellations, we hypothesized that Dice scores and CNRs are higher for 7T compared to o3T MRI scans, as assessed with one-tailed paired samples t-tests per structure. For each one-tailed test, two models were obtained. The first model ($M_1$) tested for a positive effect in support of our hypotheses, and a second model ($M_2$), tested for a negative effect in which 7T is either no different or is outperformed by the o3T MRI data. The

**Table 1. Bayes factor interpretation.**

| | Bayes Factor Interpretation | |
|---|---|---|
| $BF_{10}$ | >100 | Decisive evidence |
| *Evidence for H₁* | 30–100 | Very strong evidence |
| | 10–30 | Strong evidence |
| | 3–10 | Substantial evidence |
| | 1–3 | Anecdotal evidence |
| | 1 | No evidence |
| $BF_{01}$ | 1–0.33 | Anecdotal evidence |
| *Evidence for H₀* | 0.33–0.10 | Substantial evidence |
| | 0.10–0.03 | Strong evidence |
| | 0.03–0.01 | Very strong evidence |
| | <0.01 | Decisive evidence |

H1 = experimental hypothesis, and H0 = null hypothesis.

preferred model, which was the model for which strongest evidence was present, was then reported along with the model comparisons.

We had no hypotheses on the direction of potential changes in volumes or Anatomical distances across field strengths, and therefore conducted two-tailed paired samples t-tests. These analyses provided a single model testing for a difference either way, compared to the null hypothesis. Where appropriate, we calculated the reciprocal to determine the evidence supporting the null-hypothesis.

**Manual v semi-automated.** Similarly, when assessing manual and semi-automated parcellations within field strength (manual $o$3T v MIST 3T and manual 7T v MIST 7T), two-tailed paired samples t-tests were conducted for CNRs, volumes, and Anatomical distances, which we did not expect to differ.

The Dice score for the MIST output parcellations is comprised of a conjunction mask including only the voxels selected by both the MIST parcellation and the resampled manual conjunction mask. Therefore, Dice scores were not directly tested across parcellation methods.

**Semi-automated o3T v 7T.** $o$3T and 7T MIST parcellation Dice scores and CNRs were compared with a one-tailed paired samples t-test, under the assumption that both Dice scores and CNRs would be higher for 7T than for $o$3T, indicating that 7T is subject to fewer biases than $o$3T. Volumes and Anatomical distances were again assessed with two-tailed paired samples t-tests.

## Data sharing and accessibility statement

All anonymized data and analysis scripts are available from https://osf.io/4nrku/, under the terms of the Creative Commons Attribution License and complies with the rules of the General Data Protection Regulation (EU) 2016/679.

## Results

The MR contrasts are illustrated in Fig 1. QSM contrasts obtained from $o$3T and 7T sequences allowed for manual parcellation of the brain structures under investigation, resulting in biologically plausible 3D reconstructions (see Figs 2 and 3). As previously mentioned, the $c$3T images provided excellent in-plane resolution, though did not reasonably allow for anatomically accurate reconstructions due to the anisotropic voxel sizes. Therefore, no formal analyses were pursued for the clinical scans. All results have been averaged across hemisphere, and presented with a margin of error of <0.1%. See Table 2 for the results of the manual parcellations, and Tables 3 and 4 and Fig 4 for MIST parcellations.

### Manual parcellations: $o$3T v 7T

**Dice scores.** Dice scores for the GPe ($BF_{10}$ = 4.11), GPi ($BF_{10}$ = 4.22) and RN ($BF_{10}$ = 7.20) all reported substantial evidence in favor of 7T parcellations having a higher Dice than $o$3T. Additionally, these models were 33, 54 and 64 times (respectively) more likely than either no difference, or o3T having a higher Dice than 7T (referred to in the following sections as the alternative). For the SN ($BF_{10}$ = 1.06) and STN ($BF_{01}$ = 1.85), only anecdotal evidence was found in favor of 7T over $o$3T, which were 7 and 13 times more likely than the alternative, respectively. All winning models noted here were at least moderately more likely than the second model.

**Volumes.** When assessing for differences in volumes across field strength per structure, we found consistent anecdotal evidence for no difference for the GPe ($BF_{01}$ = 0.58), GPi ($BF_{01}$ = 0.31), RN ($BF_{01}$ = 0.47), SN ($BF_{01}$ = 0.35) and STN ($BF_{01}$ = 0.51). Substantial evidence was found for differences in volumes per rater for $o$3T parcellations for the RN ($BF_{10}$ = 3.89), and

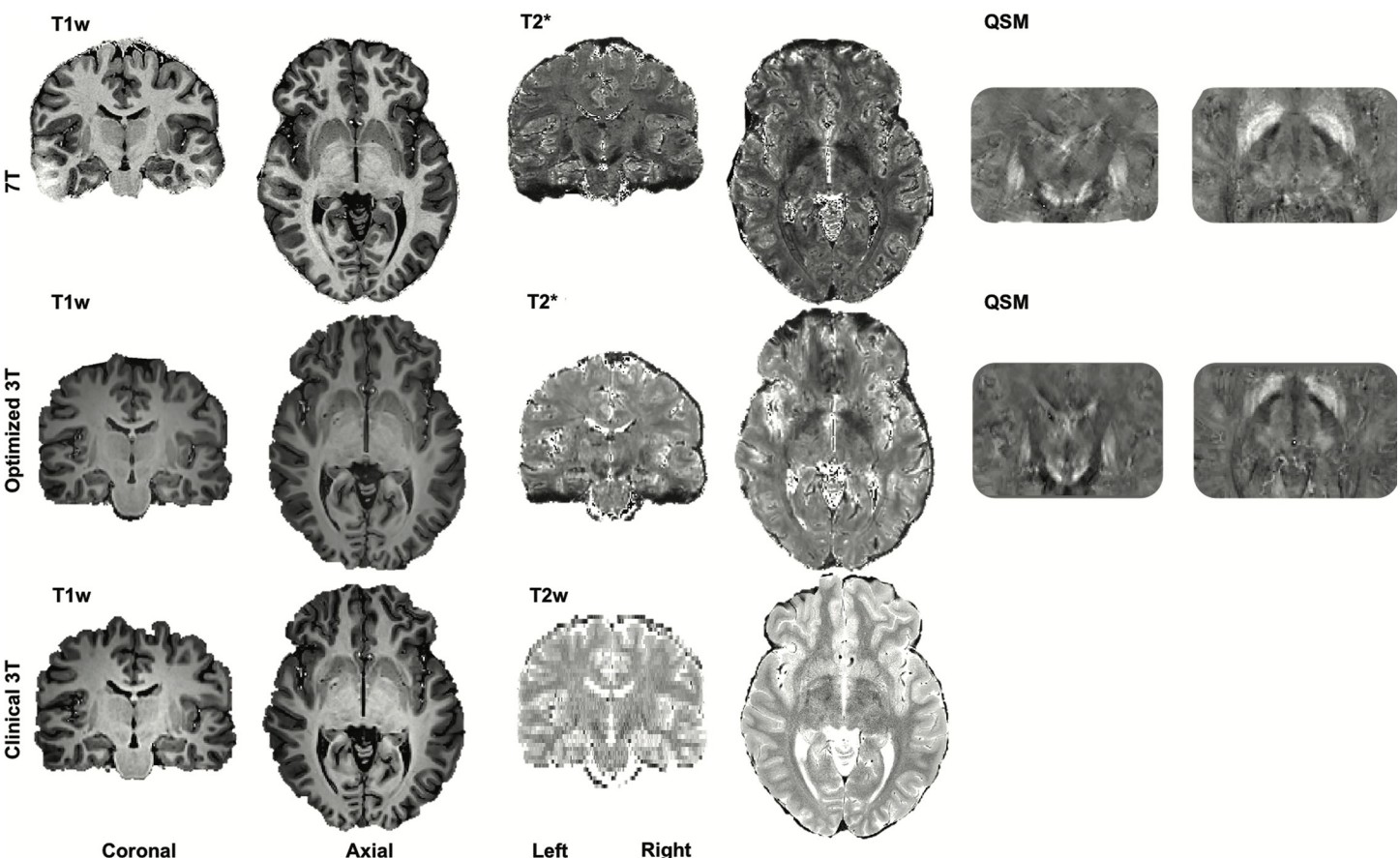

**Fig 1. Contrasts.** A single subject's 7T (T1-weighted, T2* map and QSM images), optimized 3T (o3T) (T1-weighted, T2* map and QSM images) and clinical 3T (c3T) (T1- and T2*- weighted) in the coronal and axial planes. Brightness and contrast levels were chosen to best visualize the basal ganglia.

SN ($BF_{10} = 6.72$), and at 7T, strong evidence was found for the SN ($BF_{10} = 29.87$). All other structures showed either anecdotal or no evidence for differences across raters. Surprisingly, the GPe, GPi, SN, and STN showed higher standard deviations at 7T than $o$3T. Additionally, Pearson's Rho correlation indicated that for $o$3T, Dice scores correlated with volumes for tor the GPe ($r = 0.49$), GPi ($r = 0.76$), RN ($r = 0.45$), SN ($r = 0.14$) and STN ($r = 0.61$), and at 7T for the GPe $(r = 0.61)$, GPi ($r = 0.86$), RN (r = 0.41), SN ($r = 0.80$) and STN ($r = 0.42$). This is indicative of a bias where larger structures have a higher Dice score.

**Anatomical distance.** When assessing for differences in distances across field strengths per structure, we found consistent evidence for no differences for the GPe ($BF_{01} = 0.86$, *anecdotal*), GPi ($BF_{01} = 0.89$, *anecdotal*), RN ($BF_{01} = 0.32$, *substantial*), SN ($BF_{01} = 0.31$, *substantial*), and STN ($BF_{01} = 0.34$, *anecdotal*).

## QSM CNRs

When assessing for differences in QSM CNRs for manual parcellations across field strength per structure, we found very strong evidence for higher CNRs for the STN for 7T than $o$3T ($BF_{10} = 61.75$), which was 630 times, and decisively more likely than no difference, or higher CNRs at $o$3T. However, the RN ($BF_{10} = 1.64$) and SN ($BF_{10} = 1.20$) showed only anecdotal evidence for increased CNRs at 7T than 3T, which are 12 and 8 times more likely than no differences or higher CNRs at $o$3T, respectively. For the GPi ($BF_{01} = 0.66$) anecdotal and for the GPe

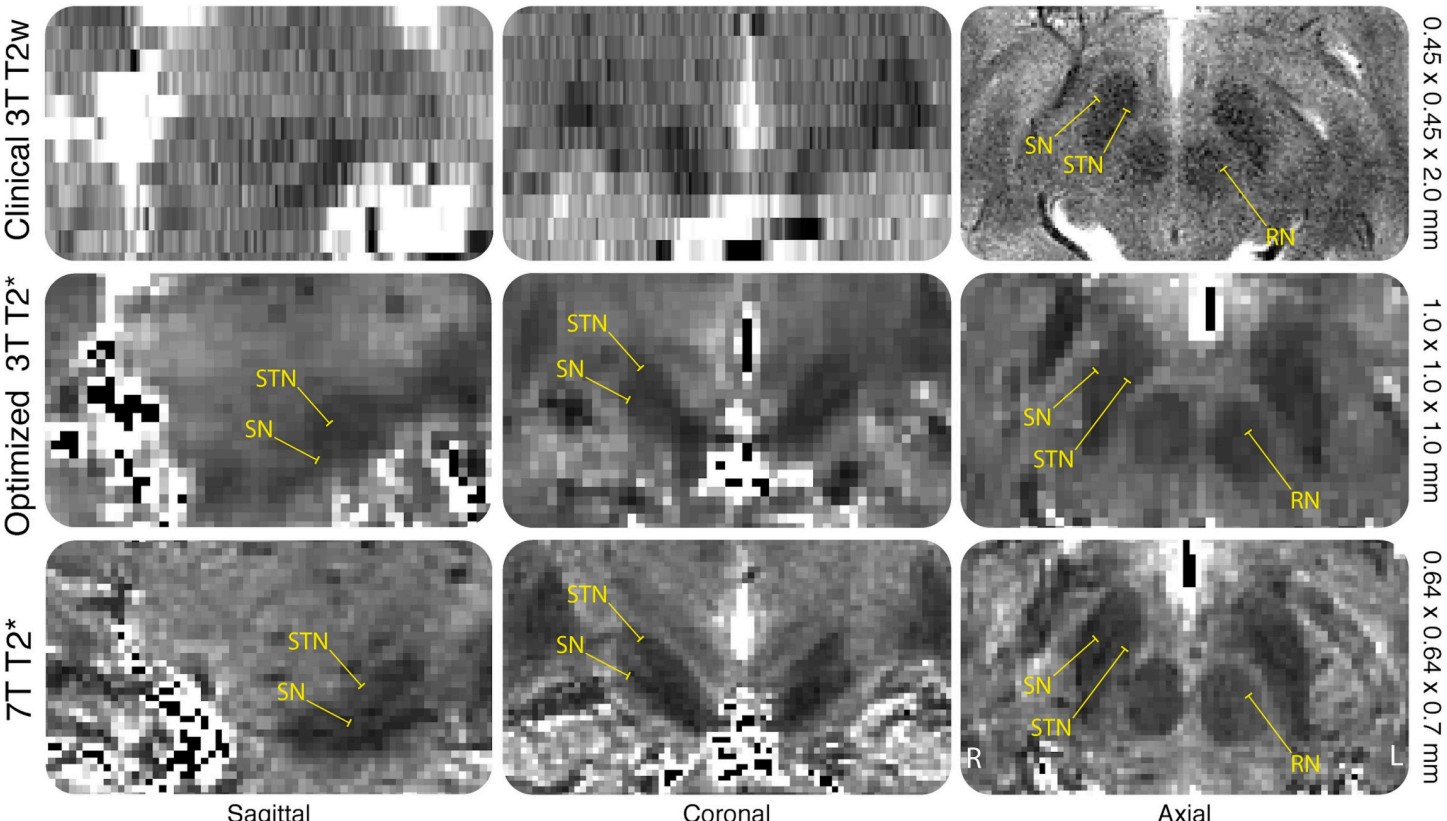

**Fig 2. Voxel sizes.** Example of a single subjects clinical 3T (c3T) T2 weighted, optimized 3T (o3T) and 7T T2* maps in the coronal, sagittal and axial planes. Voxel sizes are indicated on the right side of the figure. The RN, SN and STN are highlighted to exemplify the difficulty in identification of the nuclei in the coronal and sagittal planes for the c3T compared to the o3T and 7T due to the anisotropic voxel sizes, making 3D parcellations impossible. T2 weighted images and T2* maps are presented as they show the iron rich RN, SN and STN as hypointense structures. This was done since the c3T scan did not allow for QSM calculations, which would result in a hyperintense contrast of these brain nuclei.

($BF_{10}$ = 5.43), substantial evidence was found for increased CNR at o3T than 7T, which was 47 times more likely than higher CNRs at 7T.

## MIST parcellations

**Dice scores.** Dice scores were calculated per field strength, per structure with a one-tailed paired samples t-test for manual and semi-automated parcellations. For the GPe/i ($BF_{10}$ = 631.44), we found decisive evidence that 7T Dice scores were higher than o3T, which was 22501 times more likely than no difference, or higher Dice scores at o3T (referred to as the alternative). For the RN ($BF_{10}$ = 9.15), we found substantial evidence that 7T Dice scores were higher than o3T, which is 83 times more likely than the alternative. For the SN ($BF_{01}$ = 0.61), we found anecdotal evidence for the alternative, with increased Dice scores at o3T than 7T which is 3 times more likely than the initial hypothesis that 7T Dice scores are higher than o3T. For the STN ($BF_{10}$ = 1.04), only anecdotal evidence was found for higher Dice scores at 7T than o3T which was 7 times more likely than the alternative (see Fig 4).

**Volumes.** Two-tailed paired samples t-tests were conducted to assess differences in the volume of manual parcellations compared to the MIST output parcellations per field strength. For o3T, resulting MIST parcellation volumes were smaller than those resulting from manual parcellations for GPe/i ($BF_{10}$ = 456.18, *decisive*) and STN ($BF_{10}$ = 11.96, *strong*). For o3T SN

# optimized 3T QSM

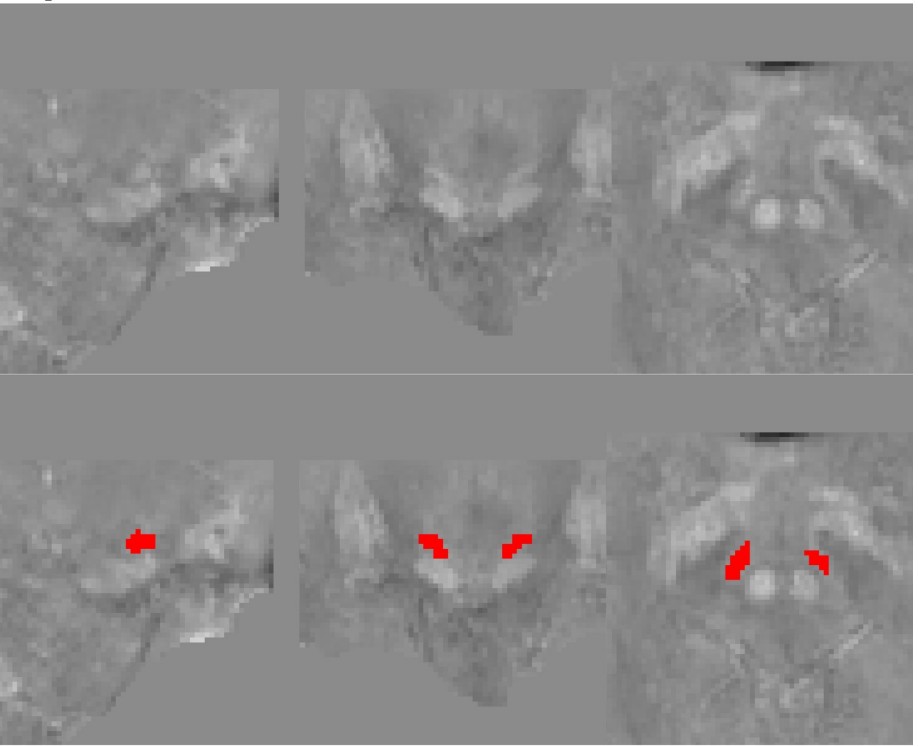

# 7T QSM

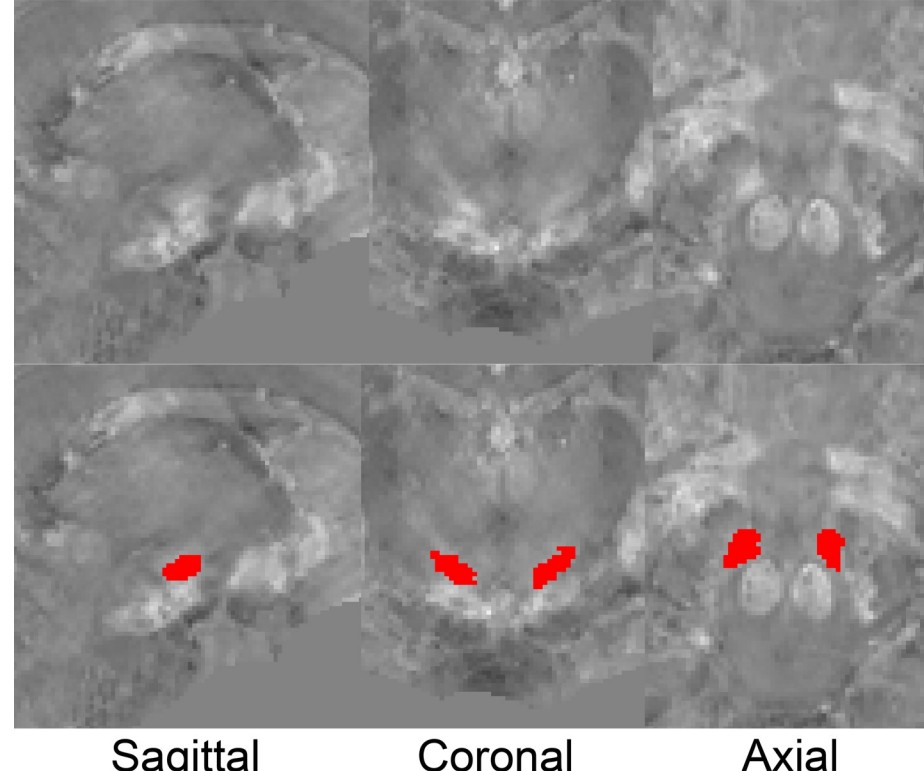

Sagittal          Coronal          Axial

**Fig 3. QSM of manual STN parcellations.** Example of a single subject parcellation of the STN on QSM images. Unlabelled and parcellated images reflect the same anatomical level in native space.

volumes larger for MIST than for manual parcellations ($BF_{10}$ = 4.10, *substantial*). For the *o*3T RN ($BF_{01}$ = 0.14, *substantial*) we have evidence for no difference. For 7T, we found evidence for no difference between manual and MIST parcellations for the RN ($BF_{01}$ = 0.33, *substantial*), SN ($BF_{01}$ = 0.74, *anecdotal*), and STN ($BF_{01}$ = 0.31, *substantial*). However, the GPe/i showed very strong evidence for increased volumes for manual parcellations than MIST ($BF_{10}$ = 36.11). Next, the volumes of MIST parcellations were compared across *o*3T and 7T. Again, for the RN ($BF_{01}$ = 0.33, *substantial*), SN ($BF_{01}$ = 0.31, *substantial*), and STN ($BF_{01}$ = 0.88, *anecdotal*), no differences in volumes were found. Finally, for the GPe/i, we found very strong evidence for increased volumes at 7T than *o*3T for MIST parcellations ($BF_{10}$ = 69.08).

**Anatomical distances.** Two-tailed paired samples t-tests were conducted to assess for differences in the anatomical distance of manual parcellations with the MIST output parcellations per field strength. For *o*3T, the GPe/i ($BF_{01}$ = 0.43), RN ($BF_{01}$ = 0.34), SN ($BF_{01}$ = 0.40), and STN ($BF_{01}$ = 0.51) all showed anecdotal evidence for no difference. For 7T, anecdotal evidence for no difference was found for the GPe/i ($BF_{01}$ = 0.37), RN (0.37) and STN (0.60), and for the SN we found anecdotal evidence for an increase in distance for MIST parcellations compared to manual ($BF_{10}$ = 1.33).

**QSM CNRs.** Two-tailed paired samples t-tests were conducted to assess differences in the CNR of manual parcellations with the MIST output parcellations per field strength. For *o3T*, the GPe/i ($BF_{10}$ = 138.13, *decisive*) and the SN ($BF_{10}$ = 3.58, *substantial*) showed evidence for increased CNRs with manual over MIST parcellations. The *o*3T RN ($BF_{01}$ = 0.32, *substantial*) and STN ($BF_{01}$ = 0.66, *anecdotal*) showed evidence for no difference in CNRs. For 7T, the GPe/i ($BF_{10}$ = 1.42) showed anecdotal evidence for higher CNRs for manual parcellations than MIST, the RN ($BF_{01}$ = 0.30) showed substantial evidence for no difference, and the SN ($BF_{10}$ =

**Table 2. Comparison of manual parcellations across optimized 3T and 7T MRI.**

| Structure | Dice scores | | | Conjunct Volumes | | | Anatomical Distance | | | QSM CNRs | | |
|---|---|---|---|---|---|---|---|---|---|---|---|---|
| | One-tailed | | | Two-tailed | | | Two-tailed | | | One-tailed | | |
| | *o3T* | *7T* | *BF* | *o3T* | *7T* | *BF* | *o3T* | *7T* | *BF* | *o3T* | *7T* | *BF* |
| **GPe** | 0.77 (0.04) | 0.81 (0.05) | $M_1$ $BF_{10}$ = 4.11 (*Mc* $BF_{10}$ = 34.26) | 1047 (127.46) | 999.52 (136.57) | $BF_{01}$ = 0.58 | 37.35 (1.67) | 36.72 (1.22) | $BF_{01}$ = 0.86 | 1.22 (0.16) | 1.09 (0.14) | $M_2$ $BF_{10}$ = 5.43 (*Mc* $BF_{10}$ = 46.65) |
| **GPi** | 0.71 (0.06) | 0.76 (0.05) | $M_1$ $BF_{10}$ = 4.22 (*Mc* $BF_{10}$ = 35.13) | 420.25 (85.71) | 415.09 (95.59) | $BF_{01}$ = 0.31 | 31.35 (2.00) | 31.95 (1.40) | $BF_{01}$ = 0.89 | 0.81 (0.16) | 0.75 (0.15) | $M_2$ $BF_{01}$ = 0.66 (*Mc* $BF_{10}$ = 3.62) |
| **RN** | 0.80 (0.06) | 0.85 (0.03) | $M_1$ $BF_{10}$ = 7.20 (*Mc* $BF_{10}$ = 63.73) | 233.40 (45.44) | 227.44 (32.59) | $BF_{01}$ = 0.47 | 8.73 (0.41) | 8.71 (0.43) | $BF_{01}$ = 0.32 | 1.92 (0.51) | 2.05 (0.42) | $M_1$ $BF_{10}$ = 1.64 (*Mc* $BF_{10}$ = 11.76) |
| **SN** | 0.78 (0.07) | 0.81 (0.03) | $M_1$ $BF_{10}$ = 1.06 (*Mc* $BF_{10}$ = 6.89) | 434.45 (75.49) | 401.39 (91.94) | $BF_{01}$ = 0.35 | 16.49 (0.83) | 16.45 (0.48) | $BF_{01}$ = 0.31 | 1.62 (0.17) | 1.78 (0.32) | $M_1$ $BF_{10}$ = 1.20 (*Mc* $BF_{10}$ = 7.87) |
| **STN** | 0.69 (0.06) | 0.74 (0.06) | $M_1$ $BF_{10}$ = 1.85 (*Mc* $BF_{10}$ = 13.59) | 88.90 (15.12) | 87.35 (23.20) | $BF_{01}$ = 0.51 | 17.93 (1.34) | 17.71 (1.47) | $BF_{01}$ = 0.34 | 1.04 (0.26) | 1.33 (0.32) | $M_1$ $BF_{10}$ = 61.75 (*Mc* $BF_{10}$ = 629.61) |

Dice scores, conjunct volumes, Anatomical distances, and QSM CNRs are averaged across hemisphere and presented as mean values and standard deviations for o3T and 7T MRI contrasts. $BF_{10}$ indicates evidence for the alternative, and $BF_{01}$ refers to evidence for the null hypothesis. Dice scores and QSM CNRs were compared using a Bayesian one-tailed paired samples t-test, where $BF_{10}$ assumes that in both cases 7T is higher than o3T (model 1 ($M_1$)), and $BF_{01}$ assumes either no difference or a decrease in 7T compared to o3T (model 2 ($M_2$)). For one-tailed paired samples t-tests, only the BF for the winning model is noted, and the likelihood ratio is calculated between the winning and losing models and is noted by 'Mc' (standing for model comparisons). Conjunct volumes and Anatomical distances were compared between 3T and 7T with two-tailed paired-samples t-tests. o3T = optimized 3T, QSM = quantitative susceptibility mapping, CNR = contrast to noise ratio, GPe = globus pallidus externa, GPi = globus pallidus interna, RN = red nucleus, SN = substantia nigra, STN = subthalamic nucleus.

**Table 3. Comparison of manual and semi-automated within and across field strength (dice scores and volumes).**

| Structure | | Dice Scores | | | Volumes | | | | | | |
|---|---|---|---|---|---|---|---|---|---|---|---|
| | | One-tailed | | | Two-tailed | | | | | | |
| | o3T [1] | 7T [1] | 3T v 7T MIST [2] BF | o3T Manual | o3T MIST | o3T MIST v Manual BF | 7T Manual | 7T MIST | 7T MIST v 7T Manual BF | o3T v 7T MIST BF |
| **GPe/i** | 0.74 (0.02) | 0.82 (0.03) | $M_1$ BF$_{10}$ = 631.44 ($Mc$ BF$_{10}$ = 22501) | 1466.20 (161.59) | 1171.91 (72.41) | BF$_{10}$ 456.18 | 1467.49 (193.23) | 1258.50 (80.45) | BF$_{10}$ 36.11 | BF$_{10}$ 69.08 |
| **RN** | 0.81 (0.06) | 0.85 (0.04) | $M_1$ BF$_{10}$ = 9.15 ($Mc$ BF$_{10}$ = 82.93) | 233.40 (45.45) | 236.55 (54.65) | BF$_{01}$ 0.14 | 236.26 (31.16) | 239.77 (39.16) | BF$_{01}$ 0.33 | BF$_{01}$ 0.33 |
| **SN** | 0.78 (0.02) | 0.75 (0.12) | $M_2$ BF$_{01}$ = 0.61 ($Mc$ BF$_{10}$ = 3.17) | 434.55 (475.36) | 503.60 (101.65) | BF$_{10}$ 4.10 | 445.47 (58.41) | 497.64 (127.78) | BF$_{01}$ 0.74 | BF$_{01}$ 0.31 |
| **STN** | 0.67 (0.08) | 0.70 (0.07) | $M_1$ BF$_{10}$ = 1.04 ($Mc$ BF$_{10}$ = 6.61) | 88.95 (15.15) | 75.25 (16.49) | BF$_{10}$ 11.96 | 90.01 (15.69) | 91.06 (24.15) | BF$_{01}$ 0.31 | BF$_{01}$ 0.88 |

[1] overlap between manual and semi-automated parcellations,

[2] 7T MIST is the preferred model.

Dice scores, conjunct volumes, Anatomical distances, and QSM CNRs are averaged across hemispheres and presented as mean values and standard deviations for o3T and 7T MRI contrasts. BF$_{10}$ indicates evidence for the alternative, and BF$_{01}$ refers to evidence for the null hypothesis. Dice scores of the agreement between manual and MIST parcellations are compared across o3T and 7T with a one-tailed paired samples t-tests where BF$_{10}$ assumes that 7T is higher than o3T (model 1 ($M_1$)), and BF$_{01}$ assumes no difference or a decrease in 7T compared to o3T (model 2 ($M_2$)). The volume, Anatomical distance and QSM CNRs are calculated for manual and MIST parcellations, and are compared both within o3T and 7T, as well as across field strength for MIST parcellations only. Volumes and Anatomical distances are compared with a two-tailed paired samples t-test. CNRs are additionally compared with a one-tailed paired samples t-tests wherein each case the BF$_{10}$ assumes that 7T CNRs are higher than o3T (model 1 ($M_1$)), and BF$_{01}$ assumes no difference or a decrease in 7T compared to o3T (model 2 ($M_2$)). For one-tailed paired samples t-tests, only the BF for the winning model is noted, and the likelihood ratio is calculated between the winning and losing models and is noted by Mc = model comparisons).

QSM = quantitative susceptibility mapping, CNR = contrast to noise ratio, GPe/i = combined globus pallidus externa and interna, RN = red nucleus, SN = substantia nigra, STN = subthalamic nucleus.

3.54, *substantial*) and STN (BF$_{10}$ = 29.89, *strong*) evidence for increased CNRs for manual parcellations than MIST. Next, CNRs of MIST parcellations were compared across o3T and 7T. The GPe/i (BF$_{01}$ = 0.70), RN (BF$_{01}$ = 0.38) and STN (BF$_{01}$ = 0.61) all showed anecdotal evidence for no difference in CNR across field strength. The SN (BF$_{10}$ = 6.02) showed substantial evidence for increased CNR at o3T than 7T, and was 51 times more likely than 7T CNRs being higher than o3T. Each winning model showed substantial evidence that it was more likely than the alternative.

## Discussion

We set out to investigate whether a clinically feasible 7T sequence can outperform optimized and clinically feasible 3T MRI protocols for the visualization of a selection of subcortical DBS or landmarks including the GPe, GPi, RN, SN, and STN. Our main findings can be summarized as follows: 1) clinical 3T MRI did not allow for accurate manual 3D parcellation of subcortical nuclei primarily due to anisotropic voxel sizes. 2) 7T outperformed optimized 3T MRI protocol for manual parcellations for larger structures (GPe, GPi and RN); 3) very strong evidence for increased QSM CNR at 7T was found for the STN when manually parcellated; 4) When using MIST for semi-automatic parcellations, Dice scores did not indicate that 7T outperformed optimized 3T.

A stated previously, clinical MRI commonly employs anisotropic voxels to increase SNR along a single direction within a shorter timeframe. As a consequence, voxels in other dimensions are elongated and suffer from partial voluming effects [57, 58]. As a result biologically plausible 3D renderings of the brain structures could not be obtained, calculations of Dice

**Table 4. Comparison of manual and semi-automated within and across field strength (anatomical distance and QSM CNRs).**

| Structure | Anatomical Distance | | | | | | | QSM CNRs | | | | | | |
|---|---|---|---|---|---|---|---|---|---|---|---|---|---|---|
| | Two-tailed | | | | | | | One-tailed | | | | | | |
| | o3T Manual | o3T MIST | o3T MIST v Manual BF | 7T Manual | 7T MIST | 7T MIST v Manual BF | o3T v 7T MIST BF | o3T Manual | o3T MIST | o3T MIST v Manual BF | 7T Manual | 7T MIST | 7T MIST v Manual BF | o3T v 7T MIST [1] BF |
| **GPe/i** | 35.61 (1.70) | 35.32 (1.38) | $BF_{01}$ 0.43 | 35.32 -1.16 | 35.45 -1.32 | $BF_{01}$ 0.37 | $BF_{01}$ 0.42 | 1.39 -0.17 | 1.21 -0.13 | $BF_{10}$ 138.13 | 1.36 (0.17) | 1.25 (0.20) | $BF_{10}$ 1.42 | $M_1$ $BF_{01}$ = 0.70 (Mc $BF_{10}$ = 3.88) |
| **RN** | 8.73 (0.40) | 8.67 (0.51) | $BF_{01}$ 0.34 | 8.77 (0.44) | 8.91 (0.69) | $BF_{01}$ 0.37 | $BF_{10}$ 1.22 | 1.89 (0.50) | 1.86 (0.47) | $BF_{01}$ 0.32 | 1.89 (0.34) | 1.89 (0.36) | $BF_{01}$ 0.3 | $M_1$ $BF_{01}$ = 0.38 (Mc $BF_{10}$ = 1.48) |
| **SN** | 16.48 (0.82) | 16.34 (0.74) | $BF_{01}$ 0.4 | 16.44 (0.45) | 17.10 (1.27) | $BF_{10}$ 1.33 | $BF_{10}$ 2.03 | 1.91 (0.18) | 1.80 (0.15) | $BF_{10}$ 3.58 | 1.94 (0.26) | 1.50 (0.35) | $BF_{10}$ 3.54 | $M_1$ $BF_{10}$ = 6.02 (Mc $BF_{10}$ = 52.39) |
| **STN** | 17.93 (1.33) | 17.55 (0.83) | $BF_{01}$ 0.51 | 17.72 (1.44) | 17.98 (1.31) | $BF_{01}$ 0.6 | $BF_{01}$ 0.6 | 1.05 (0.15) | 1.11 (0.27) | $BF_{01}$ 0.66 | 1.36 (0.19) | 1.19 (0.20) | $BF_{10}$ 29.89 | $M_1$ $BF_{01}$ = 0.61 (Mc $BF_{10}$ = 3.17) |

[1] 7T MIST is the preferred model, Two-tailed paired samples t-tests were used for these comparisons

Dice scores, conjunct volumes, Anatomical distances, and QSM CNRs are averaged across hemispheres and presented as mean values and standard deviations for o3T and 7T MRI contrasts. $BF_{10}$ indicates evidence for the alternative, and $BF_{01}$ refers to evidence for the null hypothesis. Dice scores of the agreement between manual and MIST parcellations are compared across o3T and 7T with a one-tailed paired samples t-tests where $BF_{10}$ assumes that 7T is higher than o3T (model 1 ($M_1$)), and $BF_{01}$ assumes no difference or a decrease in 7T compared to o3T (model 2 ($M_2$)). The volume, Anatomical distance and QSM CNRs are calculated for manual and MIST parcellations, and are compared both within o3T and 7T, as well as across field strength for MIST parcellations only. Volumes and Anatomical distances are compared with a two-tailed paired samples t-test. CNRs are additionally compared with a one-tailed paired samples t-tests wherein each case the $BF_{10}$ assumes that 7T CNRs are higher than o3T (model 1 ($M_1$)), and $BF_{01}$ assumes no difference or a decrease in 7T compared to o3T (model 2 ($M_2$)). For one-tailed paired samples t-tests, only the BF for the winning model is noted, and the likelihood ratio is calculated between the winning and losing models and is noted by Mc = model comparisons).

QSM = quantitative susceptibility mapping, CNR = contrast to noise ratio, GPe/i = combined globus pallidus externa and interna, RN = red nucleus, SN = substantia nigra, STN = subthalamic nucleus.

coefficients for the c3T are meaningless and were therefore not obtained. We have included the scan parameters and parcellations in a qualitative manner to illustrate the variation in target visibility across scan. We considered delineating the individual brain structures on c3T images, and performing statistical comparisons. However, c3T parcellations did not yield anatomically plausible renderings of these structures. It is conceivable that these results would have revealed a high level of interrater agreement, as well as statistically significant differences. However, given the lack of anatomical relevance of these parcellations we deemed such analyses valueless.

Arguably, identification for DBS targeting is sufficient due to the increased SNR along the axial plane, with a fast scan time of around 4 minutes [58]. However, anisotropic voxels are severely limited in their ability to portray an accurate 3D representation of the target structure and are prone to partial voluming effects [57]. Therefore, quantitative analysis of the c3T were not performed, as biological plausible 3D renderings could not be obtained. We have included the scan parameters and parcellations in a qualitative manner to illustrate the variation in target visibility across scans (see Fig 2) [59–61].

For manual parcellations, we found varying evidence in support of higher inter-rater agreement at 7T compared to the o3T MRI for the GPe/i, RN, SN and STN, which suggests that larger nuclei, and to a lesser extent, smaller nuclei have a higher visibility at increased field strengths.

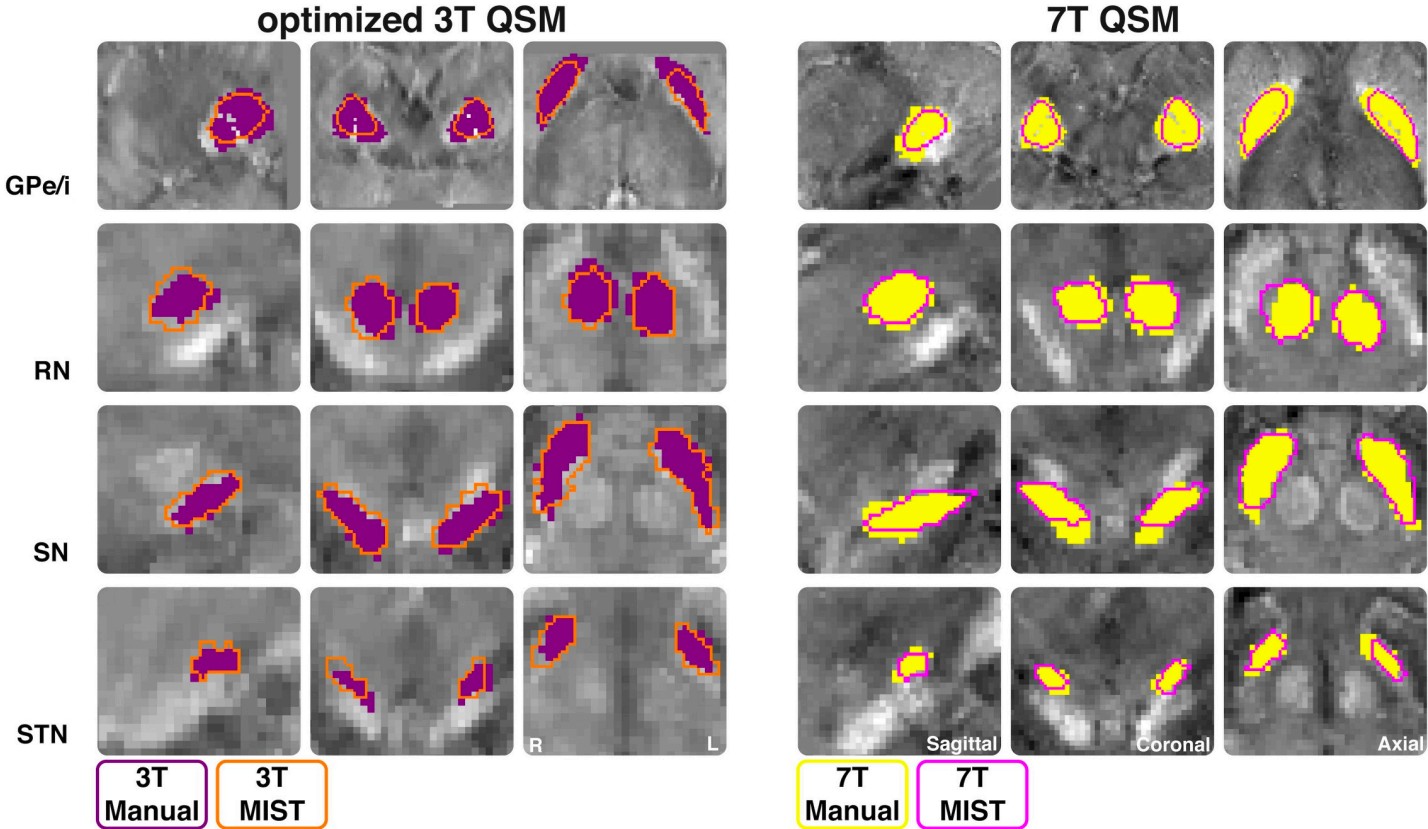

**Fig 4. MIST parcellations.** *Outline of masks for each structure manually parcellated optimized 3T (o3T) (left) and 7T (right) from a single subject. Manual masks are shown in opaque purple for o3T and yellow for 7T. MIST masks are shown as an orange outline for o3T and pink outline for 7T. Structures are shown in the coronal, sagittal, and axial planes. QSM contrasts were used for all parcellations. QSM = quantitative susceptibility mapping, GPe/i = combined globus pallidus externa and interna, RN = red nucleus, SN = substantia nigra, STN = subthalamic nucleus.*

Additionally, the volumes were smaller for all structures at 7T, which could be explained by smaller voxels and increased SNR, which counters the effects of partial voluming observed both with lower field strengths and larger voxel sizes [57]. If the changes in volume were the result of altered visibility of a specific anatomical border, this would result in a position shift of the center-of-mass. Since such a shift would be present in both hemispheres, we expected it to be reflected in a change in anatomical distance. It is important to note that it is possible that an equal but opposite effect in the other hemisphere could obscure shifts the center-of-mass. Interestingly, the CNR values were lower for 7T than $o$3T for the GPe/i. CNRs for the RN, SN, and STN were slightly higher at 7T than $o$3T, though did not result in a higher agreement between raters. This discrepancy may be explained by a raters bias with regards to prior information about size, shape and surrounding anatomy [62]. Moreover, while QSM is a quantitative measure, a strict consensus on the susceptibility values of specific structures is still lacking [63]. Additionally, smaller voxel sizes provide more precise information regarding the size and shape of structures. However, the number of voxels required to define a structure will increase, which may not directly improve the accuracy of manual parcellation. For example the STN Dice scores showed only a minor increase at 7T compared to $o$3T, despite there being evidence for increased CNR at 7T. Moreover, higher resolution allows for more freedom in where to place structural boundaries. Whereas larger voxels make labeling more reproducible, though not necessarily more anatomically accurate, we did not pursue any quantitative analyses of the $c$3T data.

For semi-automated parcellations, two sets of comparisons were computed. First, we set out to compare within field strength the differences between manual and MIST parcellations to assess for biases occurring with manual parcellations. Second, we compared MIST parcellations across field strength, as we initially did for the manual parcellations, to determine whether semi-automated protocols benefit from 7T MRI. Of note, the o3T MRI data was pre-registered to a common 1mm isotropic space to allow for computations using MIST. Similarly, to maintain the high spatial resolution but ensure compatibility, the 7T images were down sampled to 0.8 mm isotropic. We have previously demonstrated the substantial impact of voxel geometry on parcellation accuracy [57]. It is therefore possible that our results using MIST may have underestimated the effects resulting from the higher spatial resolution that can be achieved using 7T. The largely absent CNR differences indicate that differences in spatial resolution can, at least in part, explain the differences observed using MIST. Down sampling of the 7T data may have led to an underestimation of the effect of the higher spatial resolution that was obtained from 7T scanning.

Volumes for the GPe/i were very different across segmentation method. This inconsistency could be explained by the fact that manual raters parcellated the GPe and GPi separately, but were combined in MIST parcellating the GP as a single structure. Moreover, the MIST prior for the GP includes the medial medullary lamina which we did not include in our manual parcellations. Dice scores were calculated for the manual and MIST parcellations per field strength, and then compared across o3T and 7T. The RN and STN Dice scores were higher for 7T than o3T across parcellation method. Interestingly the SN Dice scores were higher for o3T than 7T. o3T MIST parcellations had smaller volumes for the GPe/i, RN and STN, and larger for the SN compared to manual parcellations. For 7T, MIST volumes were more consistent with manual parcellations. MIST parcellated volumes did not differ across field strength for the RN, SN, and STN, however the GPe/i did, which suggests larger structures may be more accurately parcellated with 7T than o3T. Generally, CNRs were higher for manual than for MIST parcellations at o3T, apart from the STN which had a higher CNR for MIST. Similarly, for 7T all CNRs were higher or equal for manual parcellations than for MIST. We found no difference in CNRs across field strength for the MIST parcellations of the palladium, RN and STN.

Interestingly, the MIST SN had higher a CNR for o3T than 7T. These findings suggest that overall, the semi-automated parcellation procedures that were applied do not appear to rely as heavily on CNR as manual parcellations and are therefore not subject to the same biases as manual parcellation. However, the SN and STN at o3T may be an exception. It may be that for smaller nuclei, semi-automated methods using lower field strengths or images with larger voxel sizes rely more on CNR for identification of structural boundaries, whereas higher field strengths or images with submillimeter resolution instead rely on the spatial information.

## Applications

It is important to consider the relevance of these findings in light of neurosurgical applications. Previous work has shown that while the visualization of the STN at 7T shows increased SNR, target localization is not necessarily improved [64–66]. We cannot conclude whether this lack of improvement can be attributed to the MRI imaging or is the result of other factors, including the variation between surgeons [66]. The current study, together with our previous findings, indicate that optimization of 3T MRI scans through the use of isotropic voxels and QSM do indeed allow for more accurate visualization of the STN [34, 57]. We developed a scan protocol optimized for its potential use in a clinical setting. Our protocol allowed the calculation of quantitative contrasts. In view of their higher sensitivity to subtle global brain changes, and

their applicability within a clinical timeframe we decided to use quantitative contrasts. We did not compare quantitative MRI to conventional weighted MRI images, and we therefore cannot conclude that quantitative MRI positively influences parcellation results. However, the theoretical benefits of removing bias, and the potential application of quantitative MRI as a biomarker argues for the calculation of quantitative MRI contrasts.

We would like to note that despite improved anatomical orientation, individual variation in the internal structure of the STN may continue to require awake testing of patients during surgery to obtain the desired clinical effect. Additionally, we have shown that an $o$3T scan can be obtained in a timeframe that is sensible within clinical practice and can account for age related increases in pathological iron deposition by using multiple and increasing echo times without superseding SAR limitations. This is a particularly important finding given the limitations of both $c$3T and 7T imaging, which include proneness to increased geometric distortions which reduce spatial accuracy and increase artefacts, B1 field inhomogeneity, power deposition, and altered specific absorption rates [2, 67]. In the MP2RAGEME, B0 inhomogeneities are automatically cancelled through the use of a ratio image [68]. Additionally, the subcortical parcellations presented in the current studies are largely dependent on grey matter contrasts, and we optimized contrast using flip angles of 4/4 instead of 7/6 degrees. Moreover, patient-related contraindications such as metal and or electronic implants, prostheses and foreign bodies, vascular or renal disorders, weight and claustrophobia can limit the potential patient population able to undergo a 7T MRI [3, 69, 70]. Thus, while our results indicate that 7T is to an extent superior to 3T, $o$3T could provide a more clinically viable option.

## Considerations

The cohort tested in this study consists of young healthy participants, and it is well known that older participants and PD patients have increased iron content in basal structures [71]. Since the effects on QSM increase with age and disease, we may underestimate the clinical relevance of these findings [34]. Moreover, the o3T consists of two separate scans, whereas the 7T acquisition includes a multi contrast scan obtained within a single session. A multi contrast scan at lower fields would have resulted in an increased scanning time, and therefore be arguably more difficult for scanning with patient populations, especially those with movement disorders. Additionally, a direct comparison between the 3T and 7T data would require co-registration to the same space involving resampling of the data. Since the outcomes of such a comparison could differ substantially dependent on the registration approach chosen, we decided not to perform such analyses. Further, it is important to note that while 3T image quality could be more closely matched with 7T MRI, the resulting protocol would have limited use for clinical application. Specifically, increasing the signal and contrast would require an more repetitions, resulting in longer acquisition times. This will further increase SAR and the impact of motion artifacts, making the potential gains in SNR and CNR arbitrary, as the scan protocol cannot be deployed clinically.

## Conclusions

We set out to test whether 7T outperformed 3T MRI in the context of target visualization for DBS surgery. We now conclude that 7T outperforms 3T protocols. $c$3T protocols do not allow the rendering of biologically plausible 3D representation of small deep brain structures, they therefore cannot provide an accurate 3D account of the surgical area. $o$3T protocols using isotropic voxels strongly improved the imaging of the surgical area, although it was still outperformed by 7T imaging. The constraints posed by the clinical applicability of the imaging protocol contributed to limitations including differences in voxel sizes, scan sequences, field

homogeneity. The results presented in the current studies should therefore be interpreted within the clinical framework, as they are not an account of the limits of 3T and 7T imaging within a research setting. Given the limited availability and compatibility restrictions in the patient population of 7T MRI systems for clinical application, our results have merit for more short-term imBprovement of clinical neuroimaging procedures for surgical purposes. Finally, the use of isotropic voxels is of great importance in these efforts, and we call for caution in the application of anisotropic voxels.

## Supporting information

**S1 File.**
(PDF)

## Author Contributions

**Conceptualization:** Bethany R. Isaacs, Martijn J. Mulder, Birte U. Forstmann, Anneke Alkemade.

**Data curation:** Bethany R. Isaacs, Martijn J. Mulder, Josephine M. Groot, Nikita van Berendonk, Nicky Lute, Pierre-Louis Bazin, Anneke Alkemade.

**Formal analysis:** Bethany R. Isaacs, Pierre-Louis Bazin, Birte U. Forstmann, Anneke Alkemade.

**Funding acquisition:** Birte U. Forstmann, Anneke Alkemade.

**Investigation:** Bethany R. Isaacs, Josephine M. Groot, Pierre-Louis Bazin, Birte U. Forstmann, Anneke Alkemade.

**Methodology:** Bethany R. Isaacs, Martijn J. Mulder, Pierre-Louis Bazin, Birte U. Forstmann, Anneke Alkemade.

**Project administration:** Bethany R. Isaacs, Nikita van Berendonk, Nicky Lute, Birte U. Forstmann, Anneke Alkemade.

**Resources:** Bethany R. Isaacs, Pierre-Louis Bazin.

**Software:** Bethany R. Isaacs, Pierre-Louis Bazin.

**Supervision:** Birte U. Forstmann, Anneke Alkemade.

**Validation:** Bethany R. Isaacs, Pierre-Louis Bazin, Birte U. Forstmann, Anneke Alkemade.

**Visualization:** Bethany R. Isaacs, Pierre-Louis Bazin, Anneke Alkemade.

**Writing – original draft:** Bethany R. Isaacs, Birte U. Forstmann, Anneke Alkemade.

**Writing – review & editing:** Martijn J. Mulder, Josephine M. Groot, Nikita van Berendonk, Nicky Lute, Pierre-Louis Bazin, Birte U. Forstmann, Anneke Alkemade.

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
