## [Decision Letter · Decision Letter 0]

3 Sep 2020

PONE-D-20-19977

3 versus 7 Tesla Magnetic Resonance Imaging for parcellations of subcortical brain structures

PLOS ONE

Dear Dr. Alkemade,

Thank you for submitting your manuscript to PLOS ONE. After careful consideration, we feel that it has merit but does not fully meet PLOS ONE’s publication criteria as it currently stands. Therefore, we invite you to submit a revised version of the manuscript that addresses the points raised during the review process.

We look forward to receiving your revised manuscript.

Kind regards,

Xi Chen

Academic Editor

PLOS ONE

Journal Requirements:

Reviewers' comments:

Reviewer's Responses to Questions

**Comments to the Author**

1. Is the manuscript technically sound, and do the data support the conclusions?

Reviewer #1: Yes

Reviewer #2: Yes

Reviewer #3: No

2. Has the statistical analysis been performed appropriately and rigorously? 

Reviewer #1: Yes

Reviewer #2: Yes

Reviewer #3: I Don't Know

3. Have the authors made all data underlying the findings in their manuscript fully available?

Reviewer #1: Yes

Reviewer #2: No

Reviewer #3: Yes

4. Is the manuscript presented in an intelligible fashion and written in standard English?

Reviewer #1: Yes

Reviewer #2: Yes

Reviewer #3: No

5. Review Comments to the Author

Reviewer #1: Isaacs et al. compared the 3 and 7 Tesla magnetic resonance imaging (MRI) in visualizing the small brain structures, with the goal of utilizing the additional imaging information afforded by the 7T MRI to eventually replace the need of behavioral testing in awake patient during deep brain stimulation (DBS) surgery. The manuscript was clearly written, and the data presentation was well-organized.

My major concern is the lack of clarity in the authors’ conclusion. They set out to ask the question whether the 7T MRI represented a better option for imaging anatomical structures such as globus pallidus (GP) and subthalamic nucleus (STN). The data seemed to suggest that the answer was “it depends.” For imaging some larger structures, the 7T did outperform, while for some other structures, it did not. However, the conclusion reached by the authors was mainly about the importance of isotropic voxels. I would recommend the authors to stay focused on the main goal of this study when it comes to the conclusions, and clearly write about whether the 7T offers additional imaging information that could potentially improve surgical procedures and patient care, and if so, to what extent.

Minor comments:

1. In the Results section of main text, under “Anatomical Distance” subtitle, when the authors reported the Bayes Factors (BFs) for GPi and STN, the interpretations seemed inconsistent with what the numbers indicated.

2. For Figure 2, please specify that the notations to the panels’ right are the voxel sizes.

Reviewer #2: Summary and overall impressions

This manuscript describes the utility of higher resolution and ultra-high field MRI for anatomical segmentations of subcortical structures with clinical applications. In particular, the manuscript compares current clinical approaches using standard 3T MRI with optimized 3T MRI and with 7T MRI and how the acquisitions affect manual and semi-automatic basal ganglia segmentations. Overall, the manuscript provides solid evidence that 7T MRI enables improvements in segmentation of subcortical brain structures, and that isotropic voxel sizes are critical for accurate subcortical segmentation. On the whole, these results are not particularly surprising, but they are important for improving clinical imaging and establishing the clinical utility of ultra-high field anatomical MRI. Certain individual statistical results also suggest weaknesses in current MRI acquisition and semi-automated processing approaches, which are important to keep in mind, particularly for clinical investigations.

The data are appropriate for this investigation and are well-described. The results are similarly interpreted appropriately. While some additional control analyses would bolster the major claims of the manuscript, and I have a few minor concerns regarding the reporting of methods and results, I can recommend this manuscript for publication if they are addressed.

Larger concerns

There are a number of uncontrolled variables that could be influencing segmentation differences between acquisitions and parcellation schemes. Two major ones—resolution and acquisition sequence—are mentioned presently, although their effects are not fully addressed.

Regarding resolution, which differs between the 7T and 3T acquisitions: a major result is the lack of CNR differences between 3T and 7T with semi-automated parcellation methods, as well as reduced segmentation consistency for smaller structures at 7T. To determine whether segmentation improvements are due to increased spatial resolution or signal contrast, it would be helpful to downsample the 7T data to the 3T resolution and compare it with both the native 7T resolution segmentations and the 3T segmentations. This downsampling would clarify the relative contributions of resolution and signal contrast.

Additionally, the manual and semi-automatic segmentations were performed on slightly different data: 3T manual segmentations were in native space, while 3T semi-automatic segmentations were on data warped to a common space. For the 7T data, manual segmentations were performed at native resolution, while semi-automatic segmentations were performed on (slightly) downsampled data. This last difference is particularly odd, given that at least one result in this manuscript is assigned to differences in resolution (3T vs. 7T small structure segmentation; Line 543) and needs controlling or further explanation.

The manuscript should address what added value the quantitative maps provide over conventional weighted images. Ideally, this would include a comparison (at 3T and at 7T) between segmentations performed by T2-weighted images (from a single echo) and T2* maps. This would be useful data for deciding the utility of quantitative maps (over conventional weighted images) in clinical contexts.

Despite a major take-home message being the importance of isotropic voxel sizes even at 3T, there is surprisingly no statistical comparison between the anisotropic (clinical) 3T scans and the isotropic (optimized) 3T scans. While not optimal, the anisotropic images do provide 3D data and could be segmented. There would likely be high variability between segmentations reflected in lower Dice scores than for the isotropic data, which would provide statistical evidence for the main conclusion of the manuscript.

Minor concerns

Regarding the presentation of statistical results, there are many comparisons made in this manuscript, each of which is appropriate. However, the tables could be difficult to read with confidence. These issues likely reflect my relative unfamiliarity with Bayesian statistics, but other readers may have similar interpretation issues, so addressing these concerns is important.

For example, noting which comparisons use one-tailed or two-tailed tests in the tables would be very helpful, as it’s currently challenging to to understand the direction/interpretation of effects at first glance. Perhaps separating or delineating the tables based on one-tailed vs. two-tailed tests would be easier to interpret.

In general, the tables are conveying a lot of statistical information but can be difficult to parse. For instance, in Table 3b, it is unclear where “Anatomical Distance” measures end and “QSM CNRs” begin. Extending the vertical separators to the highest level would help.

Further, despite relying on quantitative mapping paradigms, there is limited discussion of conventional “weighted” images vs. quantitative maps. The manuscript should discuss rationale for using maps and explain how they differ from conventional images. Incorporating a color bar in the figures (showing correspondence between image intensity and physical units) would help readers intuit the “quantitativeness” of T1 and T2* maps.

Line-specific comments

Line 142: Inconsistent reporting of TA/total acquisition time across data acquisitions in the methods (here “ms”, usually “min”). As “TA (Acquisition Time)” is ambiguous and can refer to active volume collection within a TR in a sparse fMRI design, something like “scan duration/length” or “total acquisition time” would be clearer.

Line 163: A bit more description of T1 map estimation would be helpful. In particular, T1 map estimation from the 3T data, as the o3T T1-weighted image only has one data point. Citations of previous uses of this mapping would also be beneficial. This paragraph should also be under a “Image processing” (or similarly named) section, not under “Data acquisition.”

Line 194: “QSM-maps and T1-images were used as input for MIST.” T1-weighted or T1 maps? It sounds like T1 maps but please clarify (but also see concern re: Line 163).

Line 214: Why resample to 0.8 mm? If isotropic voxel size is required, why not 0.7 mm? Additionally, right now the manual segmentation used full resolution (0.64x0.64x0.7), but MIST used 0.8 mm. To directly compare segmentation methods, statistical comparisons should be conducted on the same input data.

Line 232: Anatomical distance is computed as the center of mass between left and right hemisphere structures and is therefore sensitive to segmentation quality in each structure. Therefore, a center-of-mass difference in one hemisphere’s structure (when comparing two different scans/segmentation methods) could be masked by an equal but opposite center-of-mass difference in the other hemisphere’s structure. This is minor, and there aren’t indications that it dramatically affects the overall interpretation of results, but it should be addressed by using a control reference point (such as midline posterior commissure), or at least discussed.

Line 276: “Preferred model” could use clarification. Is it just the model with greater evidence (logarithmic distance from 1)? Tables 3a and 3b are also ambiguous in denoting the preferred model.

Line 280: For readers who are less familiar with Bayesian statistics, it would be helpful to describe differences in BF reporting between one-tailed and two-tailed tests: for one-tailed, BF > 1 suggests M1 (group 1 > group 2), while BF <1 suggests M2 (group 2 > group 1). Whereas for two-tailed, BF > 1 suggests a difference between groups without preferring one group over the other, while BF < 1 suggests no difference between groups.

Line 302: At present, the data and processing scripts are hosted on OSF but are not available without requesting access. Please make the project data and scripts public for inspection during the review process.

Table 2: QSM CNRs – GPe: o3T > 7T, but BF is described as “M_2 BF_10”. If model 2 is preferred, shouldn’t BF be reported as BF_01? (according to the caption) However, then the BF should be < 1.

Line 370: GPi (BF = 0.89) and STN (BF = 0.34) should be “anecdotal”, not “substantial” (per Bayes Factor Interpretation cut-off values in Table 1).

Line 398: Similar to Line 280, more specific statistical language would be helpful (at least for the first description of two-tailed results). For instance, could say “For o3T we found significant evidence for a difference between manual and MIST segmentations of GPe/i and STN, with both being smaller with MIST than with manual parcellations.”

Line 424: STN should be reported as BF_10, not BF_01, as per Table 3b and the present interpretation.

Table 3a: It’s ambiguous what the “preferred model” is, as the “**” annotation seems to be pointing to a statistical test, not to a specific model.

Table 3b: Same “**” preferred model issue – can’t tell which model is preferred.

Line 525: Re: segmentation method differences: address possible contributions of data resampling prior to MIST but not manual segmentation (resulting in different resolution, along with potential partial voluming effects and SNR changes).

Line 544: CNR vs. resolution could be assessed by downsampling 7T data to 3T resolution and comparing segmentations with high res 7T, o3T segmentations

Line 554: Lack of improvement at 7T could also be due to sequence differences – are there downsides to the combined T1w/T2w MP2RAGEME sequence? Does the lack of field inhomogeneity correction contribute?

Line 561: first mention of field inhomogeneity. Given that Caan et al. (2019) say that “Good B1+‐inhomogeneity correction of both MP2RAGE and MP2RAGEME data is essential for successful cortical segmentation using automated routines (Haast, Ivanov, & Uludağ, 2018)”, this manuscript should address its lack of field correction and the potential effects on subcortical segmentation.

Line 567: If further analyses are not conducted, need to add other limitations (differences in voxel sizes, sequences, field inhomogeneities, etc.).

Figure 1: Left column (T1-related) images need specific subtitles. 7T “MP2RAGE” – is it T1-weighted (UNI) or T1 map? Optimized 3T: sounds like it’s a “T1w” image, but figure suggests it’s also MP2RAGE.

Figure 3: It would be helpful to see an unsegmented QSM image next to the segmented ones so that readers can visually assess segmentation performance. If space is an issue, perhaps one side half of the brain could be unsegmented and one half could be segmented.

Reviewer #3: This manuscript has not been prepared carefully and many errors and inconsistencies in formatting need to be corrected (see below). The use of vendor-specific acronyms for pulse sequences and parameters in the Methods section is very difficult to follow without a vendor specific manual. Therefore, a full evaluation of differences in the selection of methods and parameter is not possible by the reader but should have been supplied by the authors. The c3T, o3T and 7T methods differ is so many parameters that comparing the results may not address the stated goal of the study. In the Discussion, the following statement illustrates the fundamental weakness of this comparison, “It is important to note that technically it would be possible to match the o3T scan sequence more closely to the 7T sequence. However, the resulting protocol would not meet the criteria for implementation in clinical practice due to SAR and time constraints. It is thus important to note that the improved performance of the 7T protocol cannot be exclusively attributed to differences in field strengths, but also to the constraints imposed by clinical practice.” Greater effort is necessary to adequately prepare comparable acquisition methods on the 3 T and 7 T systems to allow significant conclusion to be drawn. Also, Reference 31 demonstrates the advantage of isotropic resolution available using higher fields, like 7 T, which is a stated in the Conclusion section as a main result.

Keywords should include 7 Tesla

The chosen referencing system (names versus numbers) is not applied consistently.

Error is punctuation, e.g. missing period at the end of the sentence on line 72.

Units should be separated from numerical values (i.e. 3T, 8.2ms), but this format is not followed consistently.

On line 142, the TA units are not correct.

6. PLOS authors have the option to publish the peer review history of their article (what does this mean?). If published, this will include your full peer review and any attached files.

Reviewer #1: **Yes: **Wen Gu

Reviewer #2: No

Reviewer #3: No

---

## [Author Response · Author response to Decision Letter 0]

24 Sep 2020

We would like to thank the reviewers for carefully raising their concerns about our manuscript. We have addressed all of these comments in a pointwise manner in a separately submitted document. The page and line numbers indicated in this rebuttal correspond to the revised manuscript in which changes are reflected in bold for your convenience.

---

## [Decision Letter · Decision Letter 1]

27 Oct 2020

PONE-D-20-19977R1

3 versus 7 Tesla Magnetic Resonance Imaging for parcellations of subcortical brain structures in clinical settings

PLOS ONE

Dear Dr. Alkemade,

Thank you for submitting your manuscript to PLOS ONE. After careful consideration, we feel that it has merit but does not fully meet PLOS ONE’s publication criteria as it currently stands. Therefore, we invite you to submit a revised version of the manuscript that addresses the points raised during the review process.

We look forward to receiving your revised manuscript.

Kind regards,

Xi Chen

Academic Editor

PLOS ONE

Reviewers' comments:

Reviewer's Responses to Questions

**Comments to the Author**

1. If the authors have adequately addressed your comments raised in a previous round of review and you feel that this manuscript is now acceptable for publication, you may indicate that here to bypass the “Comments to the Author” section, enter your conflict of interest statement in the “Confidential to Editor” section, and submit your "Accept" recommendation.

Reviewer #1: All comments have been addressed

Reviewer #2: All comments have been addressed

2. Is the manuscript technically sound, and do the data support the conclusions?

Reviewer #1: Yes

Reviewer #2: Yes

3. Has the statistical analysis been performed appropriately and rigorously? 

Reviewer #1: Yes

Reviewer #2: Yes

4. Have the authors made all data underlying the findings in their manuscript fully available?

Reviewer #1: Yes

Reviewer #2: Yes

5. Is the manuscript presented in an intelligible fashion and written in standard English?

Reviewer #1: Yes

Reviewer #2: Yes

6. Review Comments to the Author

Reviewer #1: (No Response)

Reviewer #2: I am largely satisfied with the improvements in the revised manuscript and would like to thank the authors for their efforts. The authors have clarified the methods, results, and figures to a satisfactory degree. While my main critique—regarding the inconsistent acquisition methods being compared—still stands, the authors discuss the issues when relevant and have reoriented the title, abstract, introduction, and discussion to emphasize the clinical relevance of the study.

Specific comments:

Line 97: Two near identical sentences – should remove one.

Line 104: text should be changed from “we could TO run” to “we could NOT run”

Line 157: “19.53 min” should probably be “19:53 min”

Line 566: I disagree with the claim here that these results support isotropic voxels and/or QSM for improved visualization of a surgical target. The current manuscript does not test segmentation of isotropic vs. non-isotropic voxels, nor QSM vs. other modalities. If those control analyses are conducted and agree, then this claim is valid. Otherwise, this sentence should be removed.

Figure 1: Most labels have been added/corrected from the original submission; however, the Optimized 3T T1w image is missing a label (it currently reads as if it is a MP2RAGE, which it is not)

7. PLOS authors have the option to publish the peer review history of their article (what does this mean?). If published, this will include your full peer review and any attached files.

Reviewer #1: No

Reviewer #2: No

---

## [Author Response · Author response to Decision Letter 1]

27 Oct 2020

Reviewer’s question

Reviewer #2: I am largely satisfied with the improvements in the revised manuscript and would like to thank the authors for their efforts. The authors have clarified the methods, results, and figures to a satisfactory degree. While my main critique—regarding the inconsistent acquisition methods being compared—still stands, the authors discuss the issues when relevant and have reoriented the title, abstract, introduction, and discussion to emphasize the clinical relevance of the study.

Authors’ response

We would like to thank Reviewer #2 once more for meticulously reviewing our manuscript, which has had significant impact on the quality. 

Reviewer’s question

Specific comments:

Line 97: Two near identical sentences – should remove one.

Authors’ response

We have now removed the second of these two sentences. 

Reviewer’s question

Line 104: text should be changed from “we could TO run” to “we could NOT run”

Authors’ response

We have now made these changes.

Reviewer’s question

Line 157: “19.53 min” should probably be “19:53 min”

Authors’ response

We have replaced 19.53 by 19:53.

Reviewer’s question

Line 566: I disagree with the claim here that these results support isotropic voxels and/or QSM for improved visualization of a surgical target. The current manuscript does not test segmentation of isotropic vs. non-isotropic voxels, nor QSM vs. other modalities. If those control analyses are conducted and agree, then this claim is valid. Otherwise, this sentence should be removed.

Authors’ response

We have now removed the claim about the visualization of a surgical target. The text now reads:

The current study, together with our previous findings, indicate that optimization of 3T MRI scans through the use of isotropic voxels and QSM do indeed allow for more accurate visualization of the STN [34,67]. 

Reviewer’s question

Figure 1: Most labels have been added/corrected from the original submission; however, the Optimized 3T T1w image is missing a label (it currently reads as if it is a MP2RAGE, which it is not)

Authors’ response

The missing label has now been added.

---

## [Editor Report · Decision Letter 2]

9 Nov 2020

3 versus 7 Tesla Magnetic Resonance Imaging for parcellations of subcortical brain structures in clinical settings

PONE-D-20-19977R2

Dear Dr. Alkemade,

We’re pleased to inform you that your manuscript has been judged scientifically suitable for publication and will be formally accepted for publication once it meets all outstanding technical requirements.

Kind regards,

Xi Chen

Academic Editor

PLOS ONE
---

## [Editor Report · Acceptance letter]

13 Nov 2020

PONE-D-20-19977R2 

3 versus 7 Tesla Magnetic Resonance Imaging for parcellations of subcortical brain structures in clinical settings 

Dear Dr. Alkemade:

I'm pleased to inform you that your manuscript has been deemed suitable for publication in PLOS ONE. Congratulations! Your manuscript is now with our production department. 

Kind regards, 

on behalf of

Dr. Xi Chen 

Academic Editor

PLOS ONE